# Evaluation of Anal Sphincter with High Resolution Anorectal Manometry and 3D Reconstruction in Patients with Anorectal Malformation

**DOI:** 10.3390/children10061037

**Published:** 2023-06-09

**Authors:** Anna Maria Caruso, Denisia Bommarito, Vincenza Girgenti, Glenda Amato, Ugo Calabrese, Adele Figuccia, Fabio Baldanza, Francesco Grasso, Emanuela Giglione, Alessandra Casuccio, Mario Pietro Marcello Milazzo, Maria Rita Di Pace

**Affiliations:** 1Pediatric Surgical Unit, Children’s Hospital ‘G. di Cristina’, ARNAS Civico, 90100 Palermo, Italy; denisia.bommarito@arnascivico.it (D.B.); vincenza.girgenti@arnascivico.it (V.G.); glenda.amato@arnascivico.it (G.A.); ugo.calabrese@arnascivico.it (U.C.); mariopietromarcello.milazzo@arnascivico.it (M.P.M.M.); 2Pediatric Surgical Unit, Department of Health Promotion, Mother and Child Care, Internal Medicine and Medical Specialities, University of Palermo, 90100 Palermo, Italy; adelefiguccia@gmail.com (A.F.); fabio.baldanza@gmail.com (F.B.); francesco.grasso@studenti.me.it (F.G.); alessandra.casuccio@unipa.it (A.C.); mariarita.dipace@unipa.it (M.R.D.P.); 3Pediatric Surgery Division, Women’s and Children’s Health Department, University of Padua, 35100 Padua, Italy; emanuela.giglione@gmail.com

**Keywords:** anorectal malformations, high resolution anorectal manometry, fecal incontinence, bowel management

## Abstract

Background: Patients with anorectal malformation (ARM) need long-term follow-up, in order to evaluate fecal continence; the main predictors of longer-term success are the type of ARM, associated anomalies and sacral integrity. Three-Dimensional High Resolution Anorectal Manometry (3D-HRAM) gives detailed information on pressure on the anal complex profile. Our objective was to analyze anal sphincter activity in ARM patients with 3D-HRAM establishing the correlation between manometric and clinical data. Methods: Forty ARM patients were submitted to 3D-HRAM: manometric, anatomical and clinical scores were correlated with each other and with the bowel management response (BM). Results: A positive correlation between all scores and types of ARM was found: in high ARM and in patients with spinal anomalies (regardless to ARM type) lower scores were reported and even after BM they did not achieve good continence. Conclusions: 3D-HRAM gives detailed data on the functional activity of the anal sphincter complex. Our study revealed a correlation between manometric parameters and clinical outcomes, confirming spinal malformations and ARM type as the most important prognostic risk factors for a bad outcome. Specific sphincteric defects can also be explored with manometry, allowing for tailored bowel management strategies.

## 1. Introduction

Anorectal malformations (ARMs), occurring in 1:4000/5000 neonates, involve the last tract of the gastrointestinal tube and relative structures, such as the perineal body and urogenital organs [1,2,3,4,5,6,7,8]. These types of malformation fall on a wide spectrum, from simpler to more complex defects such as cloaca; therefore, functional outcomes range from good in the simplest cases to poor in the rarest and most complex situations [9,10,11,12,13,14,15,16,17,18]. The main predictors of longer-term success in fecal continence in ARM patients are the type of anorectal malformation, the presence of spinal anomalies and the sacral integrity [19,20,21,22,23,24,25,26,27], used to determine the patient’s potential for bowel control and thus their bowel management regime, such as oral medication or enemas/TAI (transanal irrigation) [28,29,30,31,32,33]. Functional assessment of the anorectum has been performed recently with High Resolution Manometry (HRM), also establishing normal values in pediatric age [34,35]. As the structural integrity and function of pelvic floor muscles plays a critical role in establishing and maintaining continence, Three-Dimensional High Resolution Anorectal Manometry (3D-HRAM) can provide an image of the pressure profile of the whole anal sphincter complex, thus allowing the evaluation of each individual muscle’s contributions to intra-anal pressure characteristics [36]. The goal of our study was to evaluate the anal sphincter function in ARM patients with high-resolution anorectal manometry and 3D reconstruction, correlating the advanced and multiplanar pressure analysis with clinical outcome and bowel management needs; as a secondary target, we evaluated how the presence of associated malformations (namely urological and spinal) can affect the prognosis.

## 2. Materials and Methods

### 2.1. Patients

A prospective analysis was conducted in our center from 2020 to 2022. A total of 40 patients up to 17 years of age, operated on for anorectal malformations, were included in the study. All patients included in this study underwent perineal surgery with either Posterior Sagittal Anorectoplasty (PSARP) or an anterior approach; we excluded patients who underwent laparoscopic surgery. We classified ARM types based on the Krickenbeck International Classification according to the type of fistula and on the basis of Wingspread classification into high, intermediate or low based on the level of the rectum in relation to the levator ani muscle on an invertogram [37,38].

Associated anomalies—above all urological and spinal—were assessed in all patients. Namely, for spinal anomalies the sacral ratio was evaluated [27,31]; other intestinal or extraintestinal anomalies were also recorded. All patients had fecal incontinence evaluated with a colonic enema and rectal ultrasound, in order to identify a megarectum in patients with retentive fecal incontinence compared to those with primary fecal incontinence, as previously described [39].

Patients with pseudo-incontinence are patients with the potential for bowel control, but who soil due to fecal impactions or because the stools move too quickly; patients with true incontinence, on the other hand, lack the ability to have a voluntary bowel movement, have poor pelvic sphincters and may have poor innervation of the rectum and sphincters in case of spinal associated anomalies.

We defined an anatomic score (reported in Table 1) for all patients, based on the anatomical perineal data: the closer to normal the patient’s perineal anatomy, the higher the score on a scale from 0 to 10.

As for clinical outcomes, we calculated the Rintala Bowel Function Score in all patients, based on seven questions, each with a possible score from 0 to 3 (except for the frequency of defecation, going from 0 to 2). We considered a total score of 18 or more as an indicator of normal bowel continence [40]. This score was evaluated before and after the introduction of bowel management (BM) with enemas or TAI. The response to BM was defined as poor, sufficient or excellent considering the difference in the Rintala score after at least 3 months of treatment with TAI; we used our previously reported TAI protocol [41].

### 2.2. Manometric Procedure

All patients underwent anorectal manometry with 3D sphincter reconstruction, as described in our previous paper [41]. The study was performed with ManoScan^®^ Anorectal High-Resolution Manometry (3D-HRAM 360/3D, Medtronic, Dublin, Ireland). Data acquisition, display, and analyses were performed with the ManoView Software.

All patients received an enema the night prior to the procedure.

The 3D-HRAM probe is a solid-state rigid tube, with 256 sensors distributed in 16 circumferences spaced along 16 axial levels. The probe has a central lumen for inflation and a luer lock at one end, through which a non-latex balloon is attached; the probe was marked to identify the anterior or posterior axis and the orientation was maintained accurately throughout, to avoid any rotation; with the patient in the left lateral position, after lubrication, the probe was gently inserted into the anal canal and slowly advanced until the upper and lower borders of the high-pressure zone were identified. The posterior marking on the probe was used by the investigator throughout the study to maintain proper orientation of the probe in relation to ventral and dorsal aspects of the anal canal. Once the probe was in place, a resting period of at least 90 s was allowed until the patients were confortable and relaxed.

During a conventional 2-dimensional (2D) high-resolution manometry, the manometry probe allows a standard visualization of the anal canal as a high-pressure zone (HPZ) at rest (anal resting pressure, ARP) and during squeeze (anal squeeze pressure, ASP), measuring its length. The high-pressure zone was identified and thirty seconds of resting pressures were obtained. The patients were then instructed to squeeze (maximum voluntary contraction), thus determining their ability to squeeze and recording the squeeze pressure characteristics (force in mmHg, reported as total pressure post increment, and its duration above or below 10 s). In order to evaluate the recto-anal inhibitory reflex (RAIR), the balloon was then rapidly inflated and deflated with a series of volumes, and a full dose-response curve was obtained. After identifying the HPZ and measuring rest and squeeze pressures, the corresponding 3D-HRAM topographic pressure view was used to obtain longitudinal and radial pressure measurements of the anal canal, both at rest and during the squeeze maneuver, in order to evaluate the dynamics of the anal channel.

With 2D- and 3D-HRAM, we evaluated the following parameters:Mean resting anal pressure;Maximum voluntary contraction during squeeze maneuver (vs. resting pressure);Sphincteric asymmetry (difference between resting and squeeze pressure above 20% between four cardinal anal segments, evaluated with 3D analysis);RAIR/rectal sensibility (recto-anal inhibitory reflex/first sensation and urge sensation)Presence of abdominal-perineal dyssynergiaGeneral aspects of pressure cylindric image

On the basis of the overall manometric results we devised a manometric score, as shown in Table 2.

Correlations between the type of ARM, presence of associated anomalies, anatomical, clinical and manometric scores were calculated; the manometric score in different types of ARM was also correlated to the BM response.

### 2.3. Statistical Analysis

Statistical analysis of the quantitative and qualitative data, including descriptive statistics, was performed for all items. The Shapiro–Wilk test was used to evaluate the normality of the distribution of the quantitative data. Continuous data were shown as mean ± standard deviation (SD). The Pearson’s chi-square test and Fisher exact test were used for frequency analysis. The univariate analysis of variance (ANOVA) was used to compare the mean differences in continuous variables between the different patient groups or different patient subgroups, and post hoc analysis with the Tukey test was used to determine whether there were pairwise intragroup differences. Moreover, a multinomial regression analysis was performed to evaluate the relationship between BM response and HRM and EO data.

Data were analyzed by IBM SPSS Software 24 version (IBM Corp., Armonk, NY, USA). All *p*-values were two-sided and *p* < 0.05 was considered statistically significant.

## 3. Results

Demographic data, types of ARM and anatomical scores are reported in Table 3.

Among male patients, 3 had recto-vesical fistulas, 5 had recto-urethral prostatic fistulas, 4 had recto-urethral bulbar fistulas and 10 had perineal fistulas; among female patients, 1 had a cloaca with a short channel, 5 had recto-vestibular fistulas, 6 had perineal fistulas and 1 had rectal atresia without a fistula.

According to the Wingspread classification, 19 pts had a high ARM, whereas 21 had a low ARM; 10 male pts also had urinary malformations (4 vesical ureteral reflux-VUR, 1 renal agenesis, 2 vesical diverticula, 1 posterior urethral valve and 2 hypospadias); 4 female pts had VUR, whereas 2 had vaginal malformation (vaginal atresia and septate uterus). Sacral and spinal malformation were reported in 11 pts (8 males and 3 females), namely tethered cord (3 pts), sacral cleft (3 pts), sacral agenesis (1 pt), hemisacrum (1 pt) and presacral mass (3 pts). In all patients with sacral anomalies the sacral ratio (SR) was less than 0.5. Colostomy was performed in 19 patients (13 males and 6 females), whereas surgery was performed in a single stage in all the others.

Correlations between type of ARM, associated anomalies, anatomical, manometric and clinical (pre and post BM) scores are reported in Table 4.

A positive correlation between anatomical score and type of ARM was found: in patients with high ARM a lower anatomical score was reported if compared with low ARM (*p* < 0.005); patients with high ARM also had lower values of Rintala and manometric scores, and even after BM they did not achieve continence score values associated with good continence, thus needing daily BM to remain clean. In patients with low ARM, the Rintala score pre-BM was better, as well as the manometric score; in these patients, the response to BM was excellent, with high values of the continence score. Regardless of ARM type, patients with spinal anomalies also showed lower values in anatomical, clinical and manometric scores and had a poor response to BM; an intermediate behavior was found in patients with urological malformations.

No correlation was found between the scores and the presence of other anomalies such as esophageal atresia, cardiac malformation or limb anomalies.

Correlations between the manometric score and response to BM in all patients, considering the type of ARM and associated anomalies, are reported in Table 5.

We found a correlation between the manometric score and response to BM: in patients with a manometric score higher than 5, the response to BM was excellent, with good continence as shown in Figure 1A,B.

Values of anal pressure, divided in the four quadrants, are reported in Table 6. All patients with ARM had values that are significantly lower than references in the literature for pediatric patients without ARM; patients with spinal anomalies, regardless of the type of ARM, showed even lower pressure values compared with other ARM patients that do not have a spinal anomaly.

In Figure 2A,B the anal channel pressure profile is shown with the differences between patients with high and low ARM, respectively: in the first case (A) the colorimetric image of the anal sphincter showed a very low pressure both at rest and during squeeze with an altered comprehensive cylindrical aspect.

## 4. Discussion

Anorectal malformations (ARM) are anomalies that cannot be corrected by surgery alone. Even for patients with excellent surgical outcomes, defecation issues can remain a lifetime experience. Some patients are very similar to normal subjects, others may have alterations of variable degrees, sometimes severe [1,2,3,4,5,6,7,8]. Associated anomalies affect up to 60–70% of ARM patients: 30% have cardiac defects, >50% urological abnormalities and 5–10% esophageal/duodenal atresia. These associated anomalies can deeply affect the prognostic outcome and influence the surgical strategy [8,9,10,11,12,13,14,15,16,17,18].

Fecal continence is associated with three main factors: sensation in the rectum, motility of the colon and sphincter control [42]: patients with ARM can have several degrees of anomalies in the muscle complex [43]. Patients with ARMs are born without an anal canal: therefore, they do not have a good sensation, perceiving only distention of the rectum (proprioception): therefore, liquid stools or soft fecal material may not be felt by the patient as the rectum is not distended.

According to Levitt and Peña, the outcome for surgical patients can be based on the anatomical classification of the ARM and the presence of sacral and/or associated spinal cord anomalies [4,17,18]. They described the morphological and functional factors responsible for constipation as megarectum, megasigmoid and denervation; fecal incontinence is correlated with an altered development of anal sphincters, impaired rectal sensation, and poor bowel motility; for these reasons, they proposed a tailored bowel management program, starting as early as possible in patients with the worst prognosis in order to keep intestinal cleaning, stooling regularity and improved quality of life [15,44].

Constipation is the most frequent morbidity encountered after the surgical repair of low ARM and is most common in patients having preserved rectosigmoid after pull-down surgery for ARMs: it occurs due to mechanical or functional reasons and its inadequate treatment can result in mega rectum/mega rectosigmoid leading to fecal impaction and overflow incontinence [17,18]. Failure to recognize or adequately treat this associated hypomotility in ARM patients can lead to significant morbidity, which we believe is largely preventable. Children treated for low ARM usually have good bowel control, but still may suffer from temporary episodes of fecal incontinence, especially when they experience diarrhea. Some 25% of all ARM children suffer from true fecal incontinence, and those are the patients who will need a constant and tailored bowel management program to keep clean: they have a hypoplastic anal sphincter, correlated to the severity of ARM, and unfortunately in some cases this can be worsened by surgery if there is no correct centering of the neoanus within the muscle complex, or if there is damage to the rectum innervation during its mobilization [11,16,17,18].

Reports of associated urological anomalies in ARM patients widely vary with reports from 18% to 85%: most series with active screening protocols report a prevalence of around 50% across all ARM types [22,23,45].

The prevalence of spinal dysraphism in patients with ARMs is about 35%, and clinically significant spinal cord tethering requiring detethering surgery is about 20%: more complex types of anorectal defects are associated with a higher prevalence of spinal dysraphism [24,25,26,27,28,29,30,31]. The presence of severe sacral abnormalities is associated with hypoplastic sphincters. If more than two sacral vertebrae are missing, or if the patient has other major sacral deformities, such as hemivertebrae and vertebral fusions, the functional outcome is worse than in patients with a normal sacrum or lesser degree of sacral maldevelopment: a sacral ratio less than 0.6 is related to a poor outcome.

Anorectal conventional manometrics allows a functional assessment of continence: pressure analysis at rest and after voluntary muscular contraction, and stimulation of rectal sensation and study of RAIR, can give good information on sphincter activity. The resting pressure is the result of a constant contraction state of the IAS at rest (85%) and partially on the EAS (15%), whereas a voluntary contraction of the EAS generates the squeezing pressure. The presence of RAIR indicates good activity of the IAS, whereas its absence can be correlated with anomalies, such as scarring, of the sphincter [35,36,37,38,39,40,41,42,43,44,45,46]. Manometric studies in children with ARM have demonstrated that low intra-anal pressures at rest-squeeze, and the absence of the recto-anal inhibitory reflex (RAIR), are associated with poor clinical outcomes and incontinence. Many patients with ARM lack the recto-anal inhibitory reflex, probably as a consequence of corrective surgery or the inborn atresia of the anal canal. An absence of RAIR is proposed to contribute to the development of constipation as it increases anal resting pressure [46,47,48,49,50]. These manometric data were well correlated with morphological studies using endoanal ultrasounds [51,52,53] and magnetic resonance (MRI) [54,55,56,57,58]. Caldaro et al. underlined how the integrity of the IAS and an adequate anal resting pressure (ARP) of 30 mm Hg were necessary to assure good continence [53]. In our previous study we demonstrated a good correlation between manometry and magnetic resonance, correlating this to the response to biofeedback and BM [59]. However, traditional water-perfused and 2D high-resolution manometric testing do not allow for full characterization of the anal canal and its individual muscle contributions: they do not provide specific information about individual components of anorectal pressure and function and so can not predict an exact prognosis. Recently, 3D high-definition anorectal manometry (3D-HRAM) has been evolved for the detailed assessment of pressure distributions in the anal canal: it has 256 sensors distributed circumferentially providing a topographic and three-dimensional (3D) pressure representation of the anal canal [36]. Well correlated with MRI and ultrasound, 3D-HRAM has been used in adults to evaluate anatomic structures and delineate the individual muscle contributions of the puborectalis (PR) muscle, internal anal sphincter (IAS), and the external anal sphincter (EAS) to intra-anal pressure characteristics [60,61,62]. In patients with ARM, a radial and longitudinal asymmetry across the anal canal and sphincter defects have been demonstrated. Elevated distal canal pressures on anorectal manometry are a primary contribution of the EAS: inadequate placement of the anal opening to the center of the EAS muscle complex affects continence, and disruption, defects, and scarring of the EAS muscle complex have been associated with low intra-anal pressures and clinically correlated with FI. During the dynamic process from rest to squeeze the ability to generate the squeeze was significantly different between controls and ARM patients, indicating a decline in the ability to voluntarily recruit the EAS muscles and generate appropriate squeeze effort, and this has been associated with altered fecal continence [50,63,64]. The PR muscle plays a key role in the maintenance of continence by the preservation of the anorectal angle at rest and by contraction and narrowing of the anorectal angle during squeeze: Dr. Alberto Peña argues that this muscle does not exist and he talks about a funnel-like muscle structure consisting of the muscle complex parallel to the rectum, the elevator muscle made out of horizontal fibers, and parasagittal fibers under the skin on each side of the anus, where the point of maximal contraction of this muscle structure could be what other authors consider the puborectalis sling. Abnormalities of the PR muscle complex disrupt its ability to modulate the anorectal angle and, therefore, compromise continence: these data have been shown in children with anorectal malformations with poor clinical outcomes and associated FI [65]. High-resolution manometry also enables a precise diagnosis of the type of dyssynergia: ARM-patients with constipation have a dyssynergic pattern type I in 90% of cases and type IV in the remaining cases [66,67]. There is a great variation in the literature regarding functional results after the repair of anorectal malformations. This is due to the fact that there is no generally agreed method to assess the bowel function of patients with anorectal malformations.

In the study of Ambartsumyan et al., manometric properties obtained with 3D-HRAM at rest were not associated with a reported predictor of fecal continence [65]. In our analysis, we confirmed the results of previous studies conducted with traditional manometry: ARM patients showed anomalies of the sphincter as lower pressure values, in particular the posterior quadrant pressures are diminished at rest and squeeze in the high ARM group vs. the low group and an altered topographic 3D view was shown; RAIR was absent in high ARM especially in cases with associated spinal anomalies. The radial and longitudinal aspect of the anal channel was different in high and low ARM. The high resolution manometric analysis showed a high-pressure zone in the posterior anal sphincter, and this area is variable according the type of ARM (less evident in high ARM in a contest of severe hypoplastic sphincter). We think that this area may correspond to what is anatomically described by Dr. Pena.

In contrast to previous 3D-HRAM studies, we found a strong correlation between the manometric results and clinical outcome, with specific regard to the type of ARM and the presence of urological and spinal anomalies. In order to simplify the comparison and correlation with the clinical continence score, we created an anatomic and a manometric score taking into account the status of the perineum post-surgery and the main parameters evaluated during manometry. Manometric results correlated well with clinical and anatomical scores; our study demonstrates that patients with high ARM and spinal anomalies show lower values of the manometric score, and that is correlated with a poor response to BM. Such correlations enhance the value of manometry, especially 3D-HRAM, as a study tool in patients with ARM, in order to provide prognostic data that correlate with the degree of continence and quality-of-life perspectives: based on manometric data, we can predict how the patient will respond to BM, and so we can tailor the BM with TAI and BFB treatment in order to ensure the maximum level of cleanliness. In collaborative patients with disruptions of the IAS, we performed a biofeedback program in order to develop the strength of the sphincter: biofeedback exercises work on specific group of muscles improving patients’ fecal control and also empowering other bowel management treatments [59]. In patients with high ARMs with severe defects of the IAS, adequate anal continence could not be achieved when IAS was absent. If the manometric analysis shows a good sphincter EAS reserve, biofeedback therapy is conducted in order to reinforce the voluntary sphincter function.

We can also confirm that the presence of associated anomalies, such as urological or spinal, negatively affects the prognosis of ARM patients. In these cases, it is mandatory to start tailored BM programs as soon as possible. This study has several limits. The main limit of our study is that we evaluated only ARM patients with fecal incontinence, and therefore we are missing a control group made up of patients with or without ARM but normally continent. The use of unvalidated scores is also one of the most important limits: we decided to create scores, both anatomical and manometric, in order to simplify as much as possible the very complex anatomical variables and manometric parameters and in order to make easier the statistical correlations between them. We decided to refer to the manometric values in the literature, as they are confirmed and standardized in several studies. Additionally, we did not correlate the manometric data with a topographic pressure view or a pure morphological study, such as an endoanal ultrasound or MRI. We chose not to include patients treated with a laparoscopic approach in this study as it is our intent to focus on those in future study: we strongly think that laparoscopically-assisted anorectal pull-through (LAARP) for high-type ARMs can reduce the amount of posterior dissection of the sphincter mechanism, required for the accurate placement of the neorectum into the muscle complex, and should therefore result in lesser disturbance of the muscle innervation, in better sphincter symmetry, and lesser irregularity and perirectal fibrosis compared to PSARP [68,69]. An aim for the future is to better clarify the role of 3D-HRAM in order to tailor the BM more and more in ARM patients.

## 5. Conclusions

Analysis by 3D-HRAM can provide information on the functional anatomy of the sphincter complex after reconstruction in patients affected by ARM. The clinical significance of this method has yet to be fully proven, but, in our opinion, our study shows a good correlation between the manometric results and the clinical outcome, additionally confirming spinal malformations and ARM type as the most important prognostic risk factors for poor outcomes. Individual patients and specific sphincteric defects can be explored with manometry, thus allowing for tailored bowel management strategies.

## Figures and Tables

**Figure 1 children-10-01037-f001:**
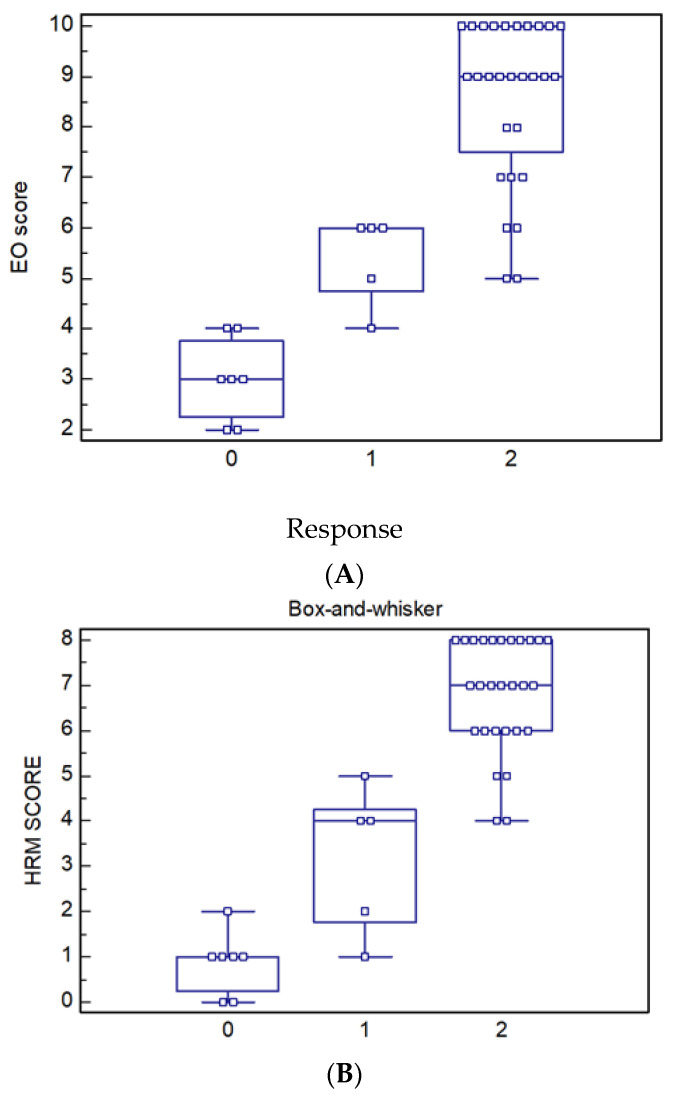
(**A**) Correlation between EO score and response to bowel management. (0: Poor; 1: discrete; 2: excellent). EO: extern anatomic objective score. (**B**) Correlation between HRM score and response to bowel management. (0: Poor; 1: discrete; 2: excellent).

**Figure 2 children-10-01037-f002:**
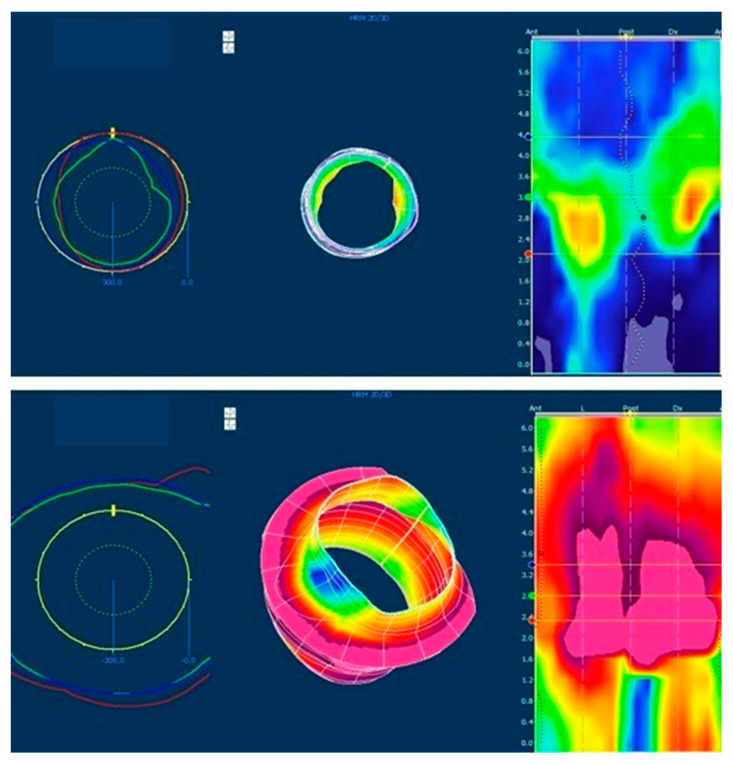
The image shows the 3D manometric reconstruction of the anal canal at rest and after squeeze (images below) in patients without (**A**) and with (**B**) sphincter defect. (**A**): patient with low Anorectal Malformation: sphincter has a good pressure value and symmetry; after voluntary contraction, the pressure increment is good for amplitude and symmetry. (**B**): patient with high Anorectal Malformation: sphincter hypotonus and asymmetry with low pressure in anterior, right and left segments at rest; during voluntary contraction (squeeze) the pressure increases but not uniformly.

**Table 1 children-10-01037-t001:** Anatomic perineal score after surgery: score 0 and score 1 were associated with a bad and good perineal aspect, respectively.

Anatomic Parameter	Score 0	Score 1
Perineal and buttock trophism	poor	good
Anal folds	ipoplasic	normal
Scrotum/fornix-anus distance	normal	short or long
Lateral deviation of neoanus	present	absent
Mucosal anal prolaps	present	absent
Perineal scar	good	ugly
Perianal lesions	present	absent
Sacral anomalies	present	absent
Intergluteal line	dysmorphic	normal
Dyscromic or fluff presacral	Present	absent

**Table 2 children-10-01037-t002:** High Resolution manometric score: score 0 and score 1 were associated with a bad and good sphincter function, respectively. HRM: High Resolution Manometry; HPZ: high pressure zone; ARP: anal resting pressure; ASP: anal squeeze pressure (reported as increment of ARP); MVC: maximal voluntary contraction; sec: seconds; RAIR: recto anal inhibitory reflex.

HRM Parameter	Score 0	Score 1
Length of HPZ (cm)	<1.5	➢ 1.5
ARP (mmHg)	<50	>50
Sphincter asymmetry (%)	➢ 20%	<20%
ASP (mmHg)	<40	➢ 40
Duration of MVC (sec)	<10	➢ 10
Dyssynergic evacuation	yes	absent
Rectal sensitivity/RAIR	Altered/absent	Normal/present
3D spatial image	Uneven cylinder	Uniform cylinder

**Table 3 children-10-01037-t003:** Demographic parameters of patients with ARM. Pts: patients; M: males; F: females; ARM: anorectal malformation; tot: total.

Variable		High ARM	Low ARM	Tot	*p*
sex	M	12	15	27	0.738
	F	7	6	13	
	tot	19	21	40	
Age (years)		8.8 ± 2.8	8.1 ± 3.1	8.5 ± 3.2	0.47
Spinal malformation		9	2	11	0.012
Urological malformation		13	3	16	<0.05
Other associated anomalies		3	2	5	0.65
Colostomy		13	0	13	<0.05

**Table 4 children-10-01037-t004:** Description of anatomical, Rintala continence (pre and post bowel management) and manometric scores according the type of ARM and associated anomalies. Pts: patients; ARM: anorectal malformation, BM: bowel management, HRM: high resolution manometry.

	High ARM	Low ARM	ARM with Uro vs. ARM without Uro	ARM with Spinal vs. ARM without Spinal	*p*
Parameter					
Anatomical score (mean)	5.05 ± 2.0	9.1 ± 1.1	5.5 ± 2.4 8.3 ± 2.1 *p* 0.00	3.8 ± 1.4 8.4 ± 1.6 *p* 0.00	<0.05
Pre BM Rintala score	5.7 ± 4.8	15.9 ± 2.6	14. 4 ± 5.3 6.6 ± 5.2 *p* 0.00	3.1 ± 3.2 14.0 ± 4.3 *p* 0.00	<0.05
HRM score	3.3 ± 2.4	7.1 ± 1.2	3.7 ± 2.5 6.3 ± 2.2 *p* 0.001	1.8 ± 1.5 6.6 ± 1.5 *p* 0.00	<0.05
Post BM Rintala score	12.3 ± 5.1	19.1 ± 1.8	13.1 ± 5.1 17.7 ± 4.1 *p* 0.003	8.9 ± 3.1 18.5 ± 2.3 *p* 0.00	<0.05
*p*	<0.05	<0.05	<0.05	<0.05	

**Table 5 children-10-01037-t005:** Correlation between HRM score and response to Bowel Management according to type of ARM and associated urological and spinal anomalies. Pts: patients; ARM: anorectal malformation, BM: bowel management, HRM: high resolution manometry; SD: standard deviation.

Type of ARM	HRM Score Mean (SD)	Response to BM			*p*
		poor	sufficient	excellent	
High ARM	19 PTS 3.32 (2.4)	7 PTS 0.86 (0.69)	3 PTS 2.67 (2.08)	9 PTS 5.44 (1.01)	<0.0005
Low ARM	21 PTS 7.14 (1.23)	0 PTS	2 4.0 (0.0)	19 PTS 7.47 (0.70)	<0.0005
ARM with urological malformation	16 PTS 3.75 (2.59)	5 PTS 0.60 (0.55)	3 PTS 3.67 (1.52)	8 PTS 5.75 (1.39)	<0.0005
ARM with spinal malformation	11 PTS 1.82 (1.53)	7 PTS 0.86 (0.69)	3 PTS 3.33 (1.15)	1 PTS 4.0 (0.0)	0.003

**Table 6 children-10-01037-t006:** A: Description of manometric parameters for all four sphincter quadrants in 3D reconstruction, in all patients and according the type of ARM and associated anomalies. Pts: patients; ARM: anorectal malformation, BM: bowel management, HRM: high resolution manometry; malf: malformation; ARP: anal resting pressure; ASP: anal squeeze pressure; ant: anterior; post: posterior; SD: standard deviation. B: Description of manometric parameters after 3D reconstruction in all and according the type of ARM and associated anomalies. Pts: patients; ARM: anorectal malformation, BM: bowel management, HRM: high resolution manometry; malf: malformation; SD: standard deviation; HPZ: high pressure zone; RAIR: recto anal inhibitory reflex.

A
Type of ARM	MeanARP mmHgMean (SD)	AntARPMean(SD)	PostARPMean (SD)	RightARPMean (SD)	LeftARPMean (SD)	MeanASPMean (SD)	AntASPMean (SD)	PostASPMean (SD)	RightASPMean (SD)	LeftASPMean (SD)
Total pts	35.6 (13.1)	32.9 (13.6)	37.2 (13.4)	33.6 (12.7)	33.6 (13.7)	78.9 (31.5)	74.1 (30.5)	84.1 (34.3)	75.8 (32.1)	53.1 (20.7)
High ARM	26.0 (9.6)	23.3 (9.7)	27.4 (10.3)	24.8 (9.7)	24.2 (10.5)	55.4 (19.7)	51.2 (19.9)	58.2 (21.2)	51.4 (21.1)	53.1(20.7)
Low ARM	44.3 (9.4)	41.8 (10.2)	46.1 (8.9)	41.6 (9.6)	42.2 (10.2)	100.1 (24.1)	94.9 (22.6)	107.4 (26.2)	97.9 (23.1)	98.9 (24.1)
ARM with urological malf	26.9 (8.9)	23.9 (8.7)	28.2 (10.2)	25.0 (8.7)	25.1 (9.2)	59.9 (26.3)	55.1 (26.2)	63.3 (27.5)	56.0 (27.7)	57.7 (27.8)
ARM with spinal malf	20.1 (5.0)	17.3 (5.3)	21.3 (6.1)	19.1 (5.7)	18.6 (7.1)	42.6 (12.1)	38.4 (12.2)	44.6 (13.6)	38.6 (14.7)	40.0 (14.3)
*p*	<0.005	<0.005	<0.005	<0.005	<0.005	<0.005	<0.005	<0.005	<0.005	<0.005
**B**
**Type of ARM**	**Length HPZ cm** **Mean (SD)**	**Asymmetry > 20%** **(n/%)**	**Duration of Contraction** **<10 s** **(n/%)**	**Dyssynergy** **(n/%)**	**RAIR Present**(**n/%)**
Total pts	1.9 (0.6)	11/40	13/40	19/40	11/40
High ARM	1.6 (0.5)	9/19	12/19	11/19	9/19
Low ARM	2.2 (0.5)	2/21	1/21	8/21	20/21
ARM with urological malf	1.7 (0.48)	7/16	10/16	11/16	8/16
ARM with spinal malf	1.4 (0.5)	9/11	10/11	6/11	1/11
*p*

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
