# Peer review of "Evaluation of Anal Sphincter with High Resolution Anorectal Manometry and 3D Reconstruction in Patients with Anorectal Malformation"

_children, 2023, doi:10.3390/children10061037_

Round 1
Reviewer 1 Report
Dear authors, this is an interesting and very well written article. I would recommend it for publication after some minor changes:
There is a mistake in the word "response" that appears under the Fig. 1A.
The authors point out that The puborectalis muscle plays a key role in maintenance of continence. However, Dr. Alberto Peña defends that this muscle does not exist. He talks about a funnel-like muscle structure consisting on the muscle complex parallel to the rectum, the elevator muscle made out of horizontal fibers, and parasagittal fibers under the skin on each side of the anus . According to Dr. Peña, the point of maximal contraction of this muscle structure could be what other authors consider the puborectalis sling. Please consider mentioning this fact in the discussion part.
Author Response
TO REVIEWER# 1: thank you for the positive comments. We really appreciated your comments and followed your suggestion as specified in the point-by-point reply:
- We have corrected the mistake under Figure 1A (response)
- We agree with you regarding Pena’s thought about puborectalis muscle: according Dr Pena, from an anatomical point of view it is not correct speak of puborectalis muscle but rather of a complex muscle structure with a point of maximal contraction considered also as puborectalis sling. The high resolution manometric analysis showed us a high-pressure zone in posterior anal sphincter, and this area is variable according the type of ARM (less evident in high ARM in a contest of severe hypoplastic sphincter. We think that this area may correspond to what is anatomically described by Dr. Pena. We added this consideration in the text on the discussion section
Reviewer 2 Report
This manuscript describes the findings of Three-Dimensional High Resolution Anorectal Manometry (3D-HRAM) in 40 patients operated for an anorectal malformations (ARM). The investigation was performed during the period 2020-22 ant the patients were up to 17 years old. Obviously, the mean age was around 8 years as it appears from Table 3?
The results of the 3D-HRAM were compared to the clinical outcomes of anatomic results, incontinence, and the results from bowel management (BM).
When it comes to the anatomic result a total of 10 parameters were investigated, some of which are relatively well-defined and others not such as e.g., “ana folds”. Who performed the investigation and registration. Were they blinded to other clinical outcomes? Was the scoring system evaluated? This is important to know as many of the 10 parameters are very subjective!
The scoring of the parameters for the HRAM – how were they chosen and were they evaluated?
Table 3. The statistical analysis does not make any sense. It is well-known that spinal and urological malformations are more common in patients with a “high ARM. It seems a peculiar that on 13 out of 19 patients with high ARM had a diverting colostomy and none in the low ARM. Did all pateints without a stoma undergo reconstruction in the neonatal period?
In Table 4 and 5 the scoring of the different parameters in relation to the characteristics of the ARM and presence of associated urological malformations are presented in addition to the score after bowel management (BM). This is very difficult to understand. Is it correct that none of the patients had undergone BM prior to the present investigation or does the values given represent those patients that had undergone a BM previously? Please clarify.
It would be interesting to know more about the bowel manage applied. Did it include any ACE-procedure? It seems that the anatomic score is a good a prediction for the outcome of BM as for the HRAM score?
The manuscript is difficult to follow, and it could benefit form at substantial shortening. The manuscript could focus upon the clinical value of HRAM . Does it ad any information to the anatomic scoring?
Minor corrections needed.
Author Response
TO REVIEWER# 2:. Thank you for your comments; we appreciated and followed your suggestions and modified our paper in accordance with them, as specified in the point-by-point reply.
- We added in table 3 the mean age for all patients
- Regarding the anatomic parameters, we didn’t use a validated score because, as far as we know, none exist. We decided to create a score considering parameters/anatomical aspects of perineal area in the postoperative period: some parameters refer to congenital anatomy of the patient according the severity of ARM, other parameters refer to postsurgical aspect. We agree with you that the anatomical evaluation may be somewhat subjective and so variable and therefore, we decided to create this easy and simplified score with only two possible variables (normal/altered) because obviously it is much easier to evaluate what is normal (compared with patients without ARM), while on the contrary, the pathological aspect can have many variables and so could be too difficult to quantify. The use of this score has made easier the statistical correlation between anatomy and manometric parameters.
- Also, for manometric score, as well as the anatomic score, we decided to create a score. In our experience and in particular, during this analysis in patients with ARM, we found an important correlation between ARM severity (perineal area altered) and manometric parameters (comparing our results with those obtained in patients with other pathologies and with those reported in the literature). An important correlation between manometric parameters and response to BM was also found. In order to perform a statistical analysis, we summarized the main manometric parameters in a score. Detailed manometric analysis was also reported in table in order to analyse difference according type of ARM and associated anomalies.
- Our data agree with the literature regarding the major incidence of spinal and urological malformation in high ARM; in Table 3 we have shown the percentage of associated anomalies in order to confirm what was said before and in order to better show (in the next tables) the correlation between spinal and urological anomalies and anatomical, clinical and manometric parameters; even if the literature is discordant on this aspect , we found that the presence of associated anomalies, especially spinal, is associated with a worse outcome independently to type of ARM.
- Our surgical approach to ARM is to operate, when possible, all low ARM in neonatal period without colostomy with a variable time interval based on ability to perform an evacuative nursing as in perineal fistula, operated about after 4-6 months.
- In this study, in order to have a sample as possible homogeneous as possible and avoid bias, we included and compared with each other only incontinent patients; all these included patients have never been subjected to BM according our protocol; some patients have been treated with enema but with poor results. We decided to correlate scores with response to BM in order to better understand which are the better prognostic parameters to consider in the prognostic outcome of ARM patients.
- Our BM protocol with transanal irrigation was described in our previous paper (ref 41)
- None of these included patients had an ACE procedure.
- Our results seem to confirm what has already been described in the literature regarding the worse outcome in patients with high ARM; using high resolution manometry, we confirmed also the alterations described on ultrasound and resonance: for the first time these alterations, showed on 3D reconstruction, have been correlate very well with clinical outcome and response to BM.
- The manuscript was shortened.
- As far as we know, a correlation between easier anatomical evaluation with score and a manometric analysis, was never reported especially considering High resolution 3D manometry: this exam provides very interesting and clear signs regarding anomalies of sphincter contraction especially during dynamic phase in squeeze manoeuvre.
Reviewer 3 Report
Dear colleagues, I sincerely thank you for your work. This is actually very important and impressive!!! I would only ask -
1- edit the introduction (there are several inconsistencies and repetitions)
2- added information about the results of irrigography (if you did - as it is usually the gold standard of study, coupled with MRI in ARM))
Accept after minor revision (corrections to minor methodological errors and text editing)
Author Response
TO REVIEWER# 3: thank you for the positive comments. We really appreciated your comments and followed your suggestion.
- All our patients, according BM protocol, were submitted to radiologic enema in order to establish if the rectum and colon are dilated: we described (as in a previous our work, ref 39) also evaluation of the rectum with sovrapubic ultrasound: without radiation we can study the presence of megarectum as sign of retentive fecal incontinence. Our future aim will be to reserve enema only for complicated cases or when the rectal ultrasound is not decisive.
- The manuscript was shortened
Reviewer 4 Report
Dear authors,
This is an important study about the evaluation of the function of the anal sphincter in patients with anorectal malformations. Any effort to improve the treatment and quality of life of these patients should be appreciated. The manuscript is well structured, but requires some improvements in terms of typographical errors.
Also, there should be no citations in the conclusions chapter, but only the conclusion of the study.
Author Response
TO REVIEWER# 4: thank you for the positive comments. We really appreciated your comments and followed your suggestion
- We removed the reference in conclusion section
Round 2
Reviewer 2 Report
Thank you for all your answers but I miss some of the comments in the manuscript. The scoring systems that use are not evaluated which is a limitation of the study. This aspect should be included in the Discussion section.
Author Response
TO REVIEWER# 2:. Thank you for your comments; we appreciated your approval regarding the modified paper; regarding your last comment, we agree with you: the use of a not validated score is one of the most important limits of this study. We decided to create scores, both anatomical and manometric, in order to simplify as much as possible, the very complex anatomical variables and manometric parameters and in order to make easier the statistical correlations between them.
We added this in the text.